# Quantifying Lumbar Foraminal Volumetric Dimensions: Normative Data and Implications for Stenosis—Part 2 of a Comprehensive Series

**DOI:** 10.3390/medsci12030034

**Published:** 2024-07-22

**Authors:** Renat Nurmukhametov, Manuel De Jesus Encarnacion Ramirez, Medet Dosanov, Abakirov Medetbek, Stepan Kudryakov, Laith Wisam Alsaed, Gennady Chmutin, Gervith Reyes Soto, Jeff Ntalaja Mukengeshay, Tshiunza Mpoyi Chérubin, Vladimir Nikolenko, Artem Gushcha, Sabino Luzzi, Andreina Rosario Rosario, Carlos Salvador Ovalle, Katherine Valenzuela Mateo, Jesus Lafuente Baraza, Juan Carlos Roa Montes de Oca, Carlos Castillo Rangel, Salman Sharif

**Affiliations:** 12nd National Clinical Centre, Federal State Budgetary Research Institution, Russian Research Center of Surgery Named after Academician B.V. Petrovsky, 109240 Moscow, Russia; renatspine20@gmail.com (R.N.); dr.medetdosanov@gmail.com (M.D.); abakirovmedetbek@gmail.com (A.M.); stepankudryakov77@gmail.com (S.K.); 2Department of Neurosurgery, Russian People’s Friendship University, 121359 Moscow, Russia; leithwisamalsaed@gmail.com (L.W.A.); neuro2009@yandex.ru (G.C.); 3Department of Head and Neck, Unidad de Neurociencias, Instituto Nacional de Cancerología, Mexico City 110411, Mexico; gervith_rs@hotmail.com; 4Neurosurgery Departament, Clinique Ngaliema, Kinshasa 3089, Democratic Republic of the Congo; drntalaja@yahoo.fr (J.N.M.); cherubin.tshiunza@neurochirurgie.fr (T.M.C.); 5I.M. Sechenov First Moscow State Medical University (Sechenov University), 119991 Moscow, Russia; vn.nikolenko@yandex.ru; 6Department of Neurosurgery, Research Center of Neurology, 125367 Moscow, Russia; 7Department of Neurosurgery, University of Pavia, 27100 Pavia, Italy; sabino.luzzi@unipv.it; 8Autonomous University of Santo Domingo (UASD), Santo Domingo 10103, Dominican Republic; andreinarosario07r@gmail.com; 9Department of Neurosurgery, National University of Mexico Hospital General, Durango 87106, Mexico; csotneurocx@outlook.com; 10Department of Spine, Central Hospital of the Armed Forces, Santo Domingo 10602, Dominican Republic; dravalenzuelacolumna@icloud.com; 11Spine Center Hospital del Mar, Sagrat Cor University Hospital, 08029 Barcelona, Spain; jlbspine@gmail.com; 12Deparment of Neurosurgery, Complejo Asistencial Universitario de Salamanca, University of Salamanca, 37008 Salamanca, Spain; juancroam@usal.es; 13Department of Neurosurgery, Servicio of the 1ro de Octubre Hospital of the Instituto de Seguridad Servicios Sociales de los Trabajadores del Estado, Mexico City 07760, Mexico; neuro_cast27@yahoo.com; 14Department of Neurosurgery, Liaqat National Hospital and Medical School, Stadium Road, Karachi 74800, Pakistan; sharifsalman73@gmail.com

**Keywords:** lumbar foraminal stenosis, lumbar spine, spine surgery

## Abstract

Introduction: Lumbar foraminal stenosis (LFS) occurs primarily due to degenerative changes in older adults, affecting the spinal foramina and leading to nerve compression. Characterized by pain, numbness, and muscle weakness, LFS arises from structural changes in discs, joints, and ligaments, further complicated by factors like inflammation and spondylolisthesis. Diagnosis combines patient history, physical examination, and imaging, while management ranges from conservative treatment to surgical intervention, underscoring the need for a tailored approach. Materials and Methods: This multicenter study, conducted over six years at a tertiary hospital, analyzed the volumetric dimensions of lumbar foramina and their correlation with nerve structures in 500 patients without lumbar pathology. Utilizing high-resolution MRI with a standardized imaging protocol, eight experienced researchers independently reviewed the images for accurate measurements. The study emphasized quality control through the calibration of measurement tools, double data entry, validation checks, and comprehensive training for researchers. To ensure reliability, interobserver and intraobserver agreements were analyzed, with statistical significance determined by kappa statistics and the Student’s *t*-test. Efforts to minimize bias included blinding observers to patient information and employing broad inclusion criteria to mitigate referral and selection biases. The methodology and findings aim to enhance the understanding of normal lumbar foramina anatomy and its implications for diagnosing and treating lumbar conditions. Results: The study’s volumetric analysis of lumbar foramina in 500 patients showed a progressive increase in foraminal volume from the L1/L2 to the L5/S1 levels, with significant enlargement at L5/S1 indicating anatomical and biomechanical complexity in the lumbar spine. Lateral asymmetry suggested further exploration. High interobserver and intraobserver agreement levels (ICC values of 0.91 and 0.95, respectively) demonstrated the reliability and reproducibility of measurements. The patient cohort comprised 58% males and 42% females, highlighting a balanced gender distribution. These findings underscore the importance of understanding foraminal volume variations for lumbar spinal health and pathology. Conclusion: Our study significantly advances spinal research by quantifying lumbar foraminal volumes, revealing a clear increase from the L1/L2 to the L5/S1 levels, indicative of the spine’s adaptation to biomechanical stresses. This provides clinicians with a precise tool to differentiate between pathological narrowing and normal variations, enhancing the detection and treatment of lumbar foraminal stenosis. Despite limitations like its cross-sectional design, the strong agreement in measurements underscores the method’s reliability, encouraging future research to further explore these findings’ clinical implications.

## 1. Introduction

Lumbar foraminal stenosis (LFS) is a condition characterized by the narrowing of the spinal foramina, leading to compression and irritation of nerve roots. This pathology is more prevalent in older adults due to degenerative changes in spinal structures such as intervertebral discs, facet joints, ligaments, and bones. These changes cause constriction in the foramina, resulting in symptoms like pain, tingling, numbness, muscle weakness, and difficulty walking, which can significantly impact the quality of life [1,2].

The anatomy of the lumbar foramen includes the vertebral body, pedicles, intervertebral disc, articular processes, ligamentum flavum, and the zygapophyseal joint. It is divided into three zones: the entrance, mid-zone, and exit zone, with further division into subcompartments reinforced by transforaminal ligaments. These ligaments play a critical role in stabilizing the foramen and facilitating the passage of the nerve root, dorsal root ganglion (DRG), and vascular structures. Notably, the 5th lumbar nerve root occupies a significant portion of the foraminal space, highlighting the complex anatomy of this region [1,3,4,5,6].

Foraminal stenosis, the primary cause of foraminal neuropathy, arises from various acquired anatomical changes within the spine, including facet joint hypertrophy, ligament thickening, bone growths, disc disorders, and osteophyte development. These changes reduce the space available for nerve roots, leading to their compression and irritation. Additionally, congenital factors can predispose individuals to this condition from an early age. Inflammation and fibrosis in the lateral recess and extraforaminal space, triggered by degenerative disorders or post-surgical scarring, can exacerbate the condition, further reducing the space for nerve roots and increasing the risk of neuropathic pain and dysfunction [7,8,9].

The multifaceted nature of foraminal neuropathy, involving both acquired and con-genital causes, underscores the complexity of its diagnosis and treatment. Healthcare professionals must consider the underlying causes to tailor treatment strategies effectively, aiming to alleviate symptoms and improve the quality of life for affected individuals. This involves a comprehensive understanding of the condition’s anatomy, pathogenesis, and contributing factors.

The pathogenesis of lumbar foraminal neuropathy is rooted in degenerative processes affecting the spine. These processes result in structural alterations of the intervertebral discs, leading to their compression and bulging. A significant outcome of these changes is the reduction in disc height, which prompts the superior articular process to shift forward and upward, causing an anterosuperior subluxation [10,11]. This disrupts the spine’s biomechanical integrity, triggers the formation of bone spurs (osteophytosis), and thickens the ligamentum flavum. Such biomechanical disruptions distribute weight unevenly across the lumbar spine, increasing stress and altering spinal segment mechanics [12,13]. Over time, this instability, combined with degenerative changes, contributes to the development of annular fissures, disc structure distortions, facet joint enlargement, and further foraminal narrowing [14].

Not all elderly individuals with spinal stenosis show symptoms, which arise primarily from inflammation in the foraminal subcompartment rather than the degree of narrowing [15]. Factors like fibrous connections and proximity to discs and joints increase inflammation risk during spinal degeneration [16,17,18]. This inflammation can cause edema, fibrin build-up, and fibrosis, leading to nerve root entrapment, exacerbated by conditions like annular tears [19,20]. Transforaminal ligaments can also cause nerve irritation and entrapment, especially with joint changes and ligament calcification [21,22]. The dorsal root ganglia, central to neuropathic pain, continue pain signals through neuropeptide and cytokine activity, sustaining pain beyond the initial inflammation [23].

Venous congestion within the spinal column initiates a series of inflammatory reactions, leading to fibrosis and increased epidural pressure, which may result in neurogenic claudication or direct neural compression [24]. Compromised blood flow from venous congestion can also lead to ischemic neuritis, contributing to symptoms associated with foraminal stenosis [25].

Spondylolisthesis, characterized by vertebral displacement, contributes to spinal instability and potential nerve compression [26]. This condition manifests in various forms, including dysplastic, isthmic, degenerative, traumatic, and pathologic spondylolisthesis, each with distinct causes ranging from congenital abnormalities and repetitive trauma to degenerative changes and bone diseases [27,28,29,30,31].

Management strategies for spondylolisthesis and foraminal neuropathy are contingent upon the severity of symptoms and the degree of anatomical changes [32]. Initial approaches typically involve conservative treatments aimed at alleviating pain and improving functionality. Physical therapy, non-steroidal anti-inflammatory drugs (NSAIDs), and lifestyle adjustments constitute the primary conservative measures [33]. In cases where these interventions fail to offer sufficient relief or in situations characterized by significant neurological deficits, surgical options such as nerve root decompression, spinal fusion, and, in certain instances, the correction of vertebral displacement may be pursued to stabilize the affected spinal segment [34].

Ligamentum flavum hypertrophy significantly contributes to the narrowing of the spinal canal and neural compression, manifesting clinically as back pain, radiculopathy, and neurogenic claudication. The pathophysiology behind this condition involves fibro-blast proliferation and the accumulation of extracellular matrix proteins, leading to increased ligament size and fibrosis [35,36].

Foraminal neuropathy presents a complex diagnostic challenge due to the non-specific nature of physical examination findings and the potential for discrepancies between clinical symptoms and imaging results. While imaging techniques such as MRI and CT myelography offer detailed insights into the anatomical changes underlying foraminal stenosis, their findings must be carefully correlated with the patient’s clinical presentation. Radiographic evidence of spinal degeneration, including disc space narrowing and facet joint hypertrophy, is common even in asymptomatic individuals, underscoring the importance of a comprehensive diagnostic approach that integrates patient history, physical examination, and the selective use of imaging and electrodiagnostic tests [37].

The grading of foraminal stenosis based on imaging findings, particularly MRI, plays a crucial role in the diagnostic process, helping to quantify the extent of anatomical changes and guide treatment decisions. However, the variability in individual anatomy and the potential for radiographic findings to overestimate or underestimate the degree of nerve compression necessitates a judicious interpretation of these results. Electrodiagnostic studies, such as electromyography (EMG), can provide additional clarity in complex cases by differentiating between peripheral and central causes of symptoms, while emerging techniques like epiduroscopy offer a more direct assessment of the inflammatory status within the epidural and foraminal spaces [38,39].

To accurately diagnose and manage leg pain resembling symptoms of foraminal neuropathy, distinguishing between various conditions is essential. Conditions such as radiculopathy, resulting from nerve root compression due to spinal stenosis or disc herniation, present sharp, radiating pain along the nerve’s course. Extraforaminal disorders, affecting nerves exiting the spine, mimic foraminal neuropathy but originate outside the spinal canal. Neurological conditions like diabetic neuropathy exhibit diffuse, mild pain and paresthesia, differentiated through electromyography (EMG). Degenerative osteoarthritis in the hip or knee, unlike neuropathic pain, worsens with joint movement without causing paresthesia. Vascular claudication, marked by leg pain during physical activity due to inadequate blood flow, shows relief upon rest, distinguishing it from neuropathic pain [40].

The objective of this article is to establish normative volumetric parameters for lumbar foramina using high-resolution MRI data. By analyzing the volumetric dimensions of the lumbar foramina, the study aims to provide a detailed understanding of normal lumbar foraminal anatomy. This information is intended to enhance the precision of diagnostic criteria and improve the management of lumbar foraminal stenosis (LFS). The findings from this comprehensive analysis seek to differentiate between pathological narrowing and normal anatomical variations, thus aiding clinicians in the early detection and personalized treatment of lumbar spinal conditions. The study also emphasizes the reliability and reproducibility of the volumetric measurements and discusses the anatomical and clinical implications of the observed variations in foraminal volumes.

## 2. Materials and Methods

### 2.1. Continuation from the Literature Review

Building upon the foundational literature review we presented in Part 1, which identified key gaps in the current understanding of lumbar foraminal stenosis (LFS), Part 2 of our series advances this knowledge by establishing normative volumetric parameters for lumbar foramina. This study aims to empirically define what is considered normal in lumbar foraminal dimensions, thereby enhancing the precision of diagnostic criteria and refining treatment strategies.

### 2.2. Study Design

This comprehensive study was planned and executed across one tertiary hospital, NCC No. 2 Federal State Budgetary Scientific Institution Russian Scientific Center, named after. acad. B.V. Petrovsky (Central Clinical Hospital Russian Academy of Sciences)—a multidisciplinary medical center aimed at providing a detailed analysis of the volumetric dimensions of lumbar foramina and their correlation with nerve structures. Conducted over a six-year period from 2017 to 2023, the research involved a sample size of 500 patients. The primary focus was on individuals exhibiting no pathological anomalies in their lumbar foramina, thereby ensuring the quality and relevance of the data collected.

### 2.3. Selection Criteria and Patient Demographics Multicenter Approach

The study was conducted in one well-established hospital center, chosen for its advanced medical imaging facilities and expertise in spinal diagnostics. This approach ensured a diverse and representative patient sample, enhancing the generalizability of the study findings.

### 2.4. Patient Cohort

A total of 500 patients equal to 5000 foramina were carefully selected based on strict inclusion criteria (Table 1). These criteria included the absence of lumbar foramina pathology, ensuring that the study’s focus remained on normal anatomical variations and their implications.

In order to maintain the integrity and specificity of the study, stringent inclusion and exclusion criteria were established:

### 2.5. Total Patients Considered

Applied Inclusion Criteria:Age ≥ 18 YearsHigh-Quality MRI ScansNo Known Spinal PathologyNo Structural Lumbar Spine Pathology
Result: **Potentially Eligible Patients**

Applied Exclusion Criteria:Known Spinal DiseasesPoor Quality ImagingAge < 18 YearsSystemic DiseasesNeurological DisordersPregnancyResult: **500 Patients Selected**

### 2.6. Data Acquisition and Imaging Protocol

#### Magnetic Resonance Imaging (MRI)

High-resolution MRI was employed as the cornerstone of our data acquisition strategy. This non-invasive imaging technique provided detailed cross-sectional images of the lumbar spine, crucial for accurate measurements of the foramina and associated nerve structures. 

A standardized imaging protocol was adopted across all centers to maintain uniformity in data acquisition. The MRI parameters were carefully selected to enhance the visualization of the lumbar spinal anatomy, with a particular focus on the foramina and nerves, ensuring optimal image quality and diagnostic accuracy.

All patients underwent imaging using a 3-T imager (Gyroscan Intera Achieva, Philips Healthcare, Eindhoven, Netherlands) with a Synergy Spine Coil (Philips Healthcare, Eindhoven, Netherlands). The patients were placed in the supine position with a cushion under both knees. T1-weighted spin-echo sagittal and axial images and T2-weighted fast spin-echo (FSE) sagittal and axial images were obtained (TR/TE, 500/15 for T1-weighted images and 3600/120 for T2-weighted images; slice thickness, 4 mm; slice gap, 0.4 mm; field of view, 32 cm for sagittal images and 16 cm for axial images; matrix, 512 × 512; flip angle, 90°; and excitations, 3).

The dataset included the disc levels L1-L2, L2-L3, L3-L4, L4-L5, and L5-S1 for each patient. The average age of the 500 patients was 37 years.

To enhance the precision in our volumetric analysis of the lumbar foramina, we utilized RadiAnt DICOM Viewer software Version: 24.0.0196, the latest version at the time of our study.

Eight experienced researchers, comprising four radiologists and four neurosurgeons, each with between 10 and 25 years of professional experience, retrospectively analyzed the magnetic resonance imaging (MRI) scans of the selected patients. To ensure the reproducibility of the findings, they independently reviewed the sagittal and axial MRI scans. The reviews were conducted blind to the patients’ clinical information to eliminate potential bias.

### 2.7. Data Analysis and Statistical Approach

#### 2.7.1. Volumetric Analysis

Utilizing detailed geometric calculations, the volume of the lumbar foramina was estimated, incorporating formulas that consider the shape of the foramina. When approximating a foramen as an elliptical cylinder, the volume V is calculated using the formula V = π × a × b × H, where a and b are the semi-major and semi-minor axes of the ellipse, respectively, and H is the depth of the foramen (distance from the entrance of the foramen to its exit in axial view).

Major Diameter (a): On the sagittal view, we measure the longest distance across the foramen from the superior to the inferior border. Minor Diameter (b): On the sagittal view, measure the shortest distance perpendicular to **a** within the foramen. Depth (H): On the axial view, measure the distance from the anterior boundary (vertebral body or intervertebral disc) to the posterior boundary (ligamentum flavum or facet joint) of the foramen.

#### 2.7.2. Statistical Methods

The relationship between the volumetric dimensions of the lumbar foramina and the nerves was examined through descriptive statistics, correlation coefficients, and regression models.

This involved measuring dimensions such as the major and minor diameters and the depth of the foramina from the MRI scans.

Statistical analyses were conducted to examine the relationship between the volumetric dimensions of the lumbar foramina and the nerves. Descriptive statistics, correlation coefficients, and regression models were utilized to interpret the data.

### 2.8. Interobserver and Intraobserver Agreement Analysis

To ensure the reliability and reproducibility of MRI assessments, an extensive evaluation of interobserver and intraobserver agreements was conducted. Eight observers, divided equally into two groups, participated in this analysis.

To evaluate the consistency of assessments among the neurosurgeons and radiologists, interobserver and intraobserver agreements were analyzed using kappa statistics. The kappa values were interpreted as follows: values less than 0.00 signify “poor” agreement; 0.00 to 0.20 indicate “slight” agreement; 0.21 to 0.40 denote “fair” agreement; 0.41 to 0.60 signify “moderate” agreement; 0.61 to 0.80 reflect “substantial” agreement; and 0.81 to 1.00 represent “almost perfect” agreement. Additionally, the Student’s *t*-test was employed to examine the clinical outcomes, with a *p*-value of less than 0.05 deemed statistically significant. For all statistical analyses, commercially available software (SPSS, version 26.0, SPSS Inc., Chicago, IL, USA) was utilized.

#### 2.8.1. Interobserver Agreement

Four observers independently assessed the MRI images to evaluate the consistency of their evaluations across different individuals. This approach allowed for a broad assessment of diagnostic agreement, reflecting the diversity of professional expertise.

#### 2.8.2. Intraobserver Agreement

The same observers re-evaluated the MRI images at a later time to assess the consistency of their assessments over time. This measure highlighted the reliability of diagnostic interpretations by individual observers.

### 2.9. Quality Control Measures

To ensure the highest standards of accuracy and reliability in our study, we implemented a series of quality control measures during both data collection and analysis. These procedures were designed to minimize errors, handle missing or outlier data appropriately, and ensure that our findings are robust and reproducible.

### 2.10. Calibration of Measurement Tools

#### Initial Calibration

Prior to data collection, all measurement tools, including imaging software (RadiAnt DICOM Viewer) and hardware (MRI scanners), were calibrated according to manufacturer specifications. This ensured that the measurements obtained were accurate and consistent across all devices.

### 2.11. Data Entry and Validation

#### Double Data Entry

To minimize errors in data entry, all information was entered independently by four team members into our database. Any discrepancies between entries were flagged for review and resolved through consensus, ensuring data accuracy. Automated validation checks were implemented to identify missing values, outliers, or data inconsistencies. These checks prompted immediate review and correction, maintaining the quality and completeness of our dataset.

### 2.12. Training and Standardization among Researchers

#### 2.12.1. Comprehensive Training

All researchers and staff involved in data collection and analysis underwent comprehensive training, emphasizing the importance of consistency and accuracy in measurements and data handling.

#### 2.12.2. Standard Operating Procedures (SOPs)

Detailed SOPs were developed and made accessible to all team members. These documents covered every aspect of data collection, entry, analysis, and quality control, serving as a reference to ensure uniform practices across the study.

#### 2.12.3. Independent Evaluation

Multiple researchers independently assessed the MRI images and mathematical formula, further minimizing the risk of subjective bias influencing the findings. Discrepancies between observers were resolved through consensus meetings, where decisions were made based on objective criteria established prior to the study.

### 2.13. Mitigating Referral and Selection Bias

#### 2.13.1. Broad Inclusion Criteria

To counteract potential referral bias, our study adopted broad inclusion criteria. This approach allowed us to capture a wide spectrum of patients, minimizing the risk of bias that could arise from analyzing a population referred from specific clinics or with particular characteristics.

#### 2.13.2. Random Sampling

Where feasible, patients were selected through random sampling from a larger pool of eligible individuals. This strategy helped to ensure that our study population was representative of the broader population of patients without lumbar foraminal stenosis or lumbar neuropathy, thereby reducing selection bias.

### 2.14. Transparency and Reproducibility

#### 2.14.1. Open Methodology

All methods and protocols were documented in detail and made publicly available. This transparency allows other researchers to understand exactly how the study was conducted, facilitating the replication of our work and the verification of our findings.

#### 2.14.2. Data Sharing

Where possible, anonymized raw data were shared in public repositories, enabling independent analysis by other researchers. This step not only supports the open science movement but also provides an additional layer of scrutiny to validate our findings.

## 3. Results

### 3.1. Volumetric Analysis of Lumbar Foramina

The volumetric analysis across 500 patients revealed notable variations in the dimensions of lumbar foramina. We observed a progressive increase in foraminal volume from the L1/L2 level to the L5/S1 level. The mean volumes at the L5/S1 level were significantly larger than those at the L1/L2 level, indicating a trend that aligns with the anatomical and biomechanical complexities of the lumbar spine (Table 2) (Figure 1) and (Figure 2). Specifically:

**L1/L2 Level Analysis:** On the right side, the mean volume commenced at 579.92 mm^3^, with a standard deviation (SD) of ±55.

The left side presented a slightly higher mean volume of 594.43 mm^3^ (SD ± 44), suggesting a mild lateral asymmetry that warrants further biomechanical exploration.

**L2/L3 Level**: 

Right Side: A mean volume of 688.22 mm^3^ was observed, with an SD of ±55.

Left Side: The mean volume increased to 715.87 mm^3^, with an SD of ±48.

**L3/L4 Level**: 

Right Side: The volume further increased to a mean of 761.70 mm^3^, with an SD of ±59.

Left Side: Similarly, an increased mean volume of 790.30 mm^3^ was noted, with an SD of ±50.

**L4/L5 Level**: Approaching the lower lumbar spine, the volumes at this level underscore the substantial enlargement of the foramina, reflecting the spine’s adaptation to biomechanical stresses.

Right Side: Here, the mean volume expanded to 787.82 mm^3^, with an SD of ±29.

Left Side: A parallel increase was seen, with the mean volume reaching 809.61 mm^3^, with an SD of ±57.

**L5/S1 Level**:

A significant volumetric increase was observed at the L5/S1 level, where the right side exhibited a mean volume of 824.24 mm^3^ (SD ± 68). 

Similarly, the left side experienced a notable elevation to a mean volume of 862.98 mm^3^ (SD ± 62), further confirming the trend of progressive volumetric increase and highlighting the potential implications for lumbar spinal health and pathology.

### 3.2. Interobserver and Intraobserver Agreement

The reliability of our volumetric measurements was assessed using kappa statistics revealing high levels of agreement:

#### 3.2.1. Interobserver Agreement

Interobserver agreement demonstrated substantial consistency, with an ICC value of 0.91, indicating excellent reliability among different observers.

#### 3.2.2. Intraobserver Agreement

Intraobserver agreement showed almost perfect agreement, with an ICC value of 0.95, highlighting the reproducibility of measurements by the same observer at different times.

### 3.3. Interobserver and Intraobserver Agreement

Ensuring reliability and consistency in our measurements, we conducted comprehensive analyses of interobserver and intraobserver agreement. To evaluate this, we employed kappa statistics to quantify the level of agreement among the eight experienced researchers, comprising four radiologists and four neurosurgeons. Each observer independently assessed the MRI images, blind to patient information, to eliminate potential biases. High interobserver agreement, indicated by an ICC value of 0.91, demonstrated substantial reliability across different observers, confirming the robustness of our measurement protocols. This aspect of our study required detailed attention beyond the summarized data presented (Table 3):

## 4. Discussion

Our study conducted a detailed volumetric analysis of lumbar foramina in 500 patients, employing MRI imaging and a novel volume calculation approach: V = π × a × b × H, where V represents the volume of the foramen, a and b are the semi-major and semi-minor axes of the elliptical cross-section, respectively, and H is the height or depth of the foramen. This methodological innovation allowed us to accurately capture the three-dimensional complexities of foraminal spaces, critical for understanding the pathophysiological and biomechanical aspects of lumbar spinal disorders.

### 4.1. Findings in Context

Our results revealed a progressive increase in foraminal volume from the L1/L2 level to the L5/S1 level, with the largest volumes observed at the L5/S1 level. This gradient of volumetric increase aligns with the anatomical and biomechanical evolution of the lumbar spine, accommodating the increased neural and vascular structures necessary for the lower limbs and supporting the greater mechanical loads experienced by the lower lumbar segments.

Comparing our findings with the existing literature reveals a general consensus on the variability of lumbar foraminal dimensions, attributed to both congenital factors and adaptive responses to biomechanical stresses. However, the precise quantification of these volumes and their progressive increase across lumbar levels has been less frequently documented, underscoring the contribution of our study to the field.

Previous studies, such as Stephens et al. (1991), utilized a mold technique to study foraminal shape and area, reporting an average foraminal area of 101.6 mm^2^ (range 40–160 mm^2^) and average foraminal height of 14.9 mm (range 10–19 mm). While valuable, their reliance on two-dimensional casts may not fully capture the three-dimensional complexities of foraminal spaces [40,41,42,43].

Chen et al. and Torun et al. explored foraminal dimensions using molds and digital calipers, respectively. These studies highlight variability in methodologies across research on lumbar foramina. Our volumetric analysis using the formula V = π × a × b × H offers a more comprehensive approach that accurately captures the three-dimensional anatomy of lumbar foramina [44,45].

Schlegel et al. and Shin et al. used CT scans and significant software for their analyses, aiming to provide precise measurements of foraminal areas. Our method’s simplicity and direct focus on volumetric assessment provide a unique advantage in evaluating the entire volume of the foramen, rather than just area or linear dimensions [46,47]. While these studies provided valuable insights, our method distinguishes itself by its simplicity and direct focus on volumetric assessment. Instead of merely measuring two-dimensional aspects like area or linear distances, our approach captures the entire three-dimensional volume of the foramen [48,49,50]. This volumetric perspective offers a more comprehensive understanding of the foraminal space, crucial for assessing the spatial accommodation of nerve roots and surrounding structures [51,52].

Cho et al. applied MRI scans to model the foraminal space as an ellipse, using mathematical formulations to deduce foraminal dimensions before and after ALIF surgery [6,53]. This approach bears similarities to ours, emphasizing the significance of three-dimensional assessment for understanding lumbar foramen anatomy and changes post-intervention. However, our study’s contribution lies in its systematic volumetric analysis across multiple spinal levels in a large patient cohort, offering broader insights into the normal and pathological conditions of lumbar foramina.

### 4.2. Anatomical and Clinical Implications

The anatomical significance of our findings extends beyond mere measurement. The progressive enlargement of foraminal volumes towards the lower lumbar spine reflects the necessity for accommodating larger nerve roots and adapting to biomechanical stresses that increase with descending lumbar levels. Clinically, these insights are vital for diagnosing and managing lumbar foraminal stenosis (Figure 3). Understanding normal volumetric ranges and variations can help clinicians identify pathological deviations indicative of foraminal narrowing, facilitating early intervention and personalized treatment approaches [6,27].

### 4.3. Integrating Volumetric Analysis into Clinical Practice

Our volumetric analysis, grounded in a precise and replicable formula, offers a framework for integrating quantitative imaging assessments into the diagnostic process for lumbar spinal disorders. By establishing normative data for lumbar foraminal volumes, our study paves the way for developing diagnostic criteria and thresholds that can be applied in clinical settings to differentiate normal anatomical variations from clinically significant stenosis (Figure 4) [54,55,56].

MRI is a very useful tool in evaluating lumbar foraminal stenosis. However, there have been relatively few reports on the reliability of MRI-based grading or classification of lumbar foraminal stenosis and its ability to predict surgical outcomes. Grading systems based on the degree of epidural fat obliteration, according to locations of epidural fat obliteration in four quadrants of the intervertebral foramen, and more recently, based on the type of stenosis, amount of fat obliteration, and presence of nerve root compression, have been proposed. However, these systems have not made clinical correlations, such as operative indications in relation to MRI grade [57,58,59].

Right-handedness, which is predominant in the population, may lead to asymmetrical loading and stress distributions on the lumbar spine due to repetitive motions and postural habits favoring the right side. This asymmetric stress could accelerate degenerative changes in spinal structures, including the intervertebral discs, facet joints, and ligaments, on the dominant side, potentially leading to disparities in foramen size between the left and right sides [60,61].

Degenerative processes such as disc dehydration, facet joint osteoarthritis, and ligamentum flavum thickening directly contribute to lumbar foraminal stenosis (LFS), characterized by narrowing of the spinal foramina. The progressive nature of these degenerative changes, exacerbated by the biomechanical demands of right-handed activities, could result in a more pronounced reduction in foramen size on one side of the lumbar spine compared to the other [62].

The balance between maintaining spinal stability and allowing for nerve root mobility is further complicated in right-handed individuals by the tendency for repeated unilateral movements, potentially leading to uneven wear and tear on the spinal components. These factors collectively contribute to observed differences in foramen size, underscoring the importance of considering handedness and biomechanical stresses in the assessment and management of lumbar spinal conditions [63].

Frédéric Khiami et al. introduced an innovative approach for measuring lumbar foraminal volumes, demonstrating the utility and reproducibility of CT scans combined with VitreaCore^®^ software version 6.5.9 for volumetric calculations [64,65]. This study focuses on healthy subjects, using precise CT imaging to develop and validate a new method that may enhance diagnostic assessments in clinical settings. Comparatively, our MRI-based study analyzes lumbar foraminal volumes across multiple levels from L1/L2 to L5/S1 in 500 healthy subjects, showcasing a progressive increase in volumes indicative of the biomechanical complexities of the lumbar spine (Table 4). This MRI-based analysis offers detailed insights into the soft tissue components within the spinal foramina, a critical aspect for comprehensive diagnostic evaluations in complex spinal pathologies [66,67].

### 4.4. Limitations

#### 4.4.1. Cross-Sectional Study Design

One of the primary limitations is the cross-sectional nature of our research, which limits our ability to capture and analyze the dynamic changes in foraminal volume over time. Longitudinal changes, particularly in relation to the progression of lumbar foraminal stenosis and its impact on patient outcomes, remain unexplored in our study framework.

#### 4.4.2. Imaging Quality Variability

Although we have taken rigorous steps to standardize imaging procedures, variability in imaging quality, attributable to different machines and operator techniques, may have influenced the precision of our volumetric measurements. This variability poses a challenge in ensuring the absolute accuracy of the calculated foraminal volumes.

#### 4.4.3. Population Specificity

Our study population, while diverse, may not fully capture the range of anatomical variations present across broader demographics. The findings derived from our specific cohort may not be directly applicable to all patient populations, particularly those with unique spinal anatomies or underlying health conditions that could affect lumbar foraminal volumes.

### 4.5. Future Directions

#### 4.5.1. Longitudinal Studies

Future research should focus on longitudinal studies to track changes in lumbar foraminal volumes over time. This approach would provide insights into the progres-sion of lumbar foraminal stenosis (LFS) and its relationship with degenerative changes in the spine. Longitudinal data can help identify early indicators of pathological changes and predict the onset of symptoms, enabling proactive management strategies.

#### 4.5.2. Enhanced Imaging Protocols

Advancements in imaging technology and the development of standardized imaging protocols could mitigate the impact of variability in imaging quality. Future studies should explore the use of high-resolution imaging techniques and artificial intelligence algorithms to enhance the precision and reliability of volumetric measurements.

#### 4.5.3. Diverse Population Studies

Expanding research to include a wider and more diverse population base would enhance the generalizability of the findings. Comparative studies across different ethnicities, age groups, and patients with varying degrees of spinal pathology would help validate the applicability of our volumetric measurement formula and its correlation with clinical outcomes.

#### 4.5.4. Clinical Correlation Analysis

A critical area for future research is the exploration of the relationship between quantified volumetric changes in lumbar foramina and clinical outcomes in patients. This includes assessing the impact of foraminal volume changes on symptom severity, quality of life, and response to treatment in patients with lumbar foraminal stenosis.

#### 4.5.5. Validation of Measurement Formula

Further research is needed to validate and refine the formula used for calculating foraminal volumes. Comparative studies using different volumetric measurement techniques could help establish the accuracy and reliability of our formula, potentially setting a new standard for quantitative assessments of spinal anatomy.

### 4.6. Looking Ahead

Clinical Applications in Part 3: As we conclude Part 2 of this series, we have laid the groundwork with normative volumetric data that provides a new benchmark for understanding lumbar foraminal dimensions. These insights not only enrich our diagnostic toolkit but also refine our therapeutic approaches to lumbar foraminal stenosis. In the forthcoming Part 3, we will pivot to the clinical implications of these findings, focusing squarely on the clinical presentation and management strategies for lumbar foraminal stenosis. 

This next installment will delve into how our data translate into practical, clinical applications, enhancing patient outcomes through targeted interventions. We will explore a range of therapeutic modalities, from conservative management to surgical innovations, and discuss their efficacy in the light of our volumetric analysis.

## 5. Conclusions

We achieved a novel quantification of the lumbar foramina’s volumetric dimensions across a significant patient cohort. Our results indicated a clear trend of increasing foraminal volume from the L1/L2 to the L5/S1 levels, highlighting the spine’s adaptive mechanisms to biomechanical demands and its capacity to accommodate the nerve roots’ spatial requirements, particularly in the lower lumbar segments. The consistency of our volumetric measurements, reinforced by strong interobserver and intraobserver agreements, underscores the method’s reliability and reproducibility.

The clinical ramifications of our findings are significant. By establishing quantifiable norms for lumbar foraminal volumes, our research offers clinicians a precise tool for distinguishing pathological foraminal narrowing from normal anatomical variations. This is a critical step forward in the early detection and personalized treatment of lumbar foraminal stenosis, aiming to markedly improve patient care.

Acknowledging the study’s limitations, including its cross-sectional design and the variability in imaging quality, we advocate for future research to pursue longitudinal studies, utilize imaging technologies and mathematical formulas, and explore the clinical outcomes’ correlation with volumetric changes.

## Figures and Tables

**Figure 1 medsci-12-00034-f001:**
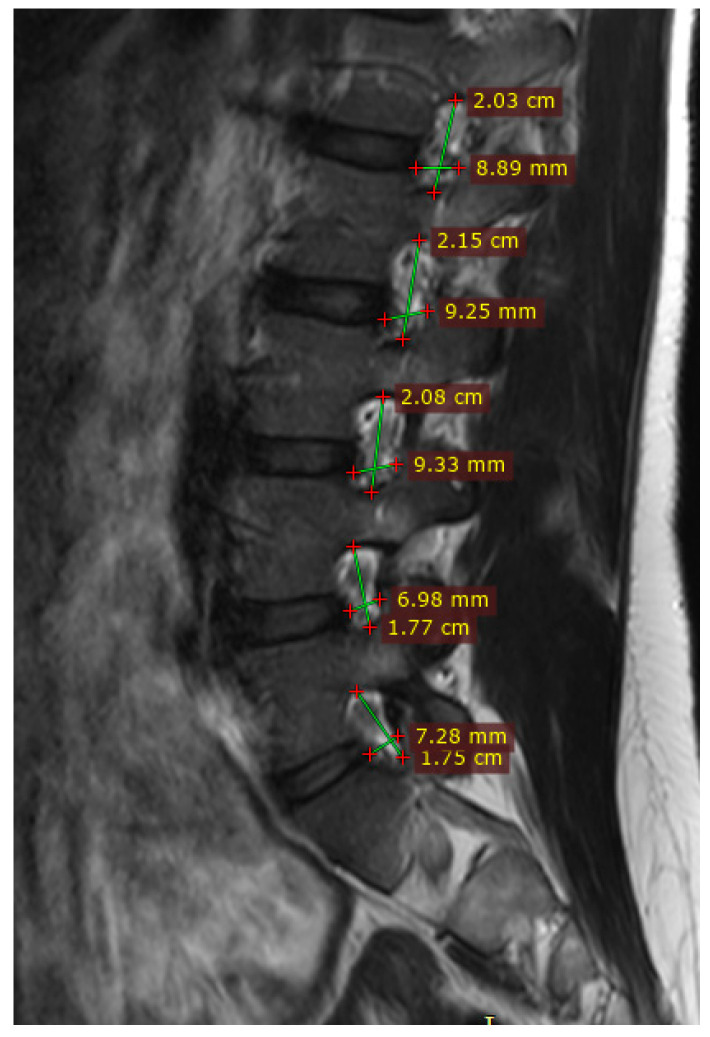
Measurements of magnetic resonance imaging parameters: sagittal view. the measurements lines of major longitudinal and shortest distance perpendicular minor size of formanen.

**Figure 2 medsci-12-00034-f002:**
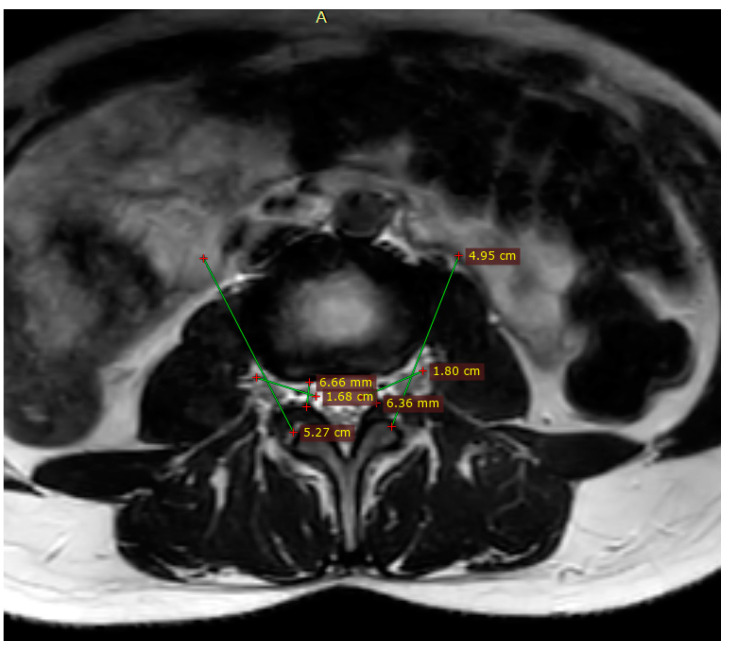
Measurements of deep of foramen in axial view magnetic resonance imaging parameters .measure the distance from the anterior boundary (vertebral body or intervertebral disc) to the posterior boundary (ligamentum flavum or facet joint) of the foramen.

**Figure 3 medsci-12-00034-f003:**
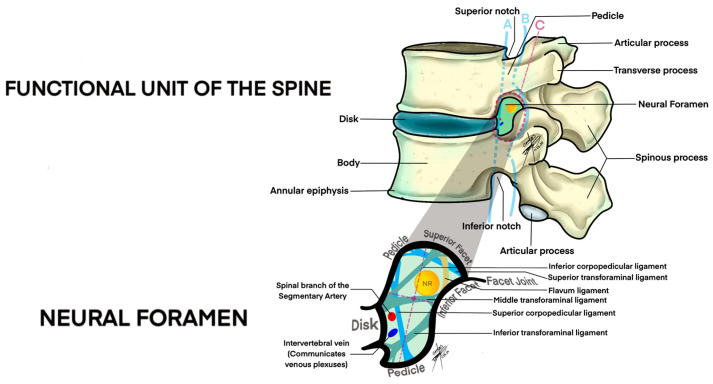
Lateral anatomical view of lumbar foraminal space. A; anterior limit of foramen, B: posterior limit of foramen, C: area total of foramen including the superior and inferior limits.

**Figure 4 medsci-12-00034-f004:**
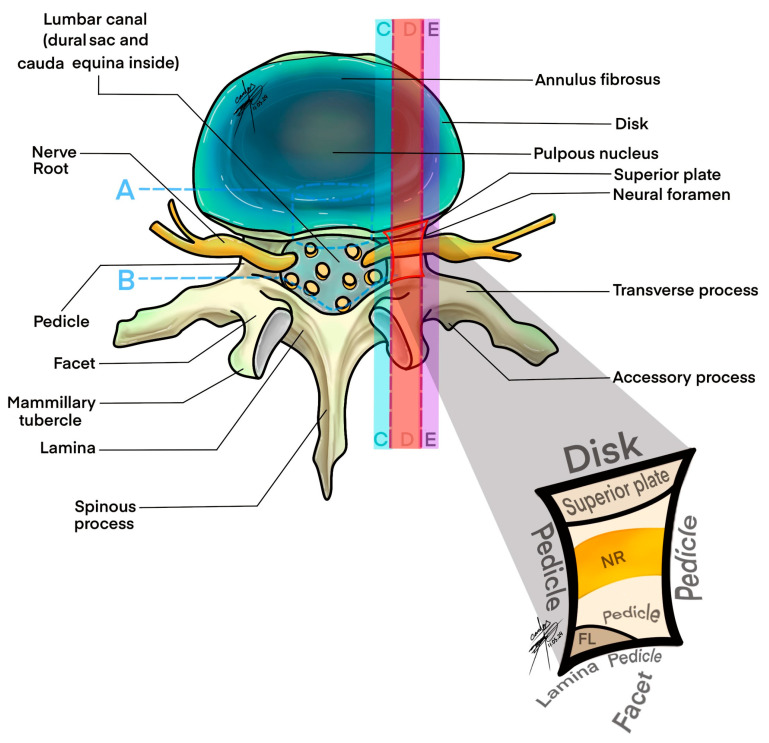
Axial view of lumbar foraminal space. A: anterior limit of foramen, B: posterior limit of foramen, C: Intraforaminal, D: foraminal, E: extraforaminal.

**Table 1 medsci-12-00034-t001:** Basic Statistical Information of Patient Cohort.

Statistic	Value
Maximum Age	65 years
Minimum Age	18 years
Mean Age	37 years
Gender	
- Male	58% (290)
- Female	42% (210)

**Table 2 medsci-12-00034-t002:** Volumetric measurement comparative table.

Lumbar Level	Segment	Mean Volume (mm^3^)	SD (mm^3^)	Number
L1/L2	Right	579.92	±55	500
L1/L2	Left	594.43	±44	500
L2/L3	Right	688.22	±55	500
L2/L3	Left	715.87	±48	500
L3/L4	Right	761.70	±59	500
L3/L4	Left	790.30	±50	500
L4/L5	Right	787.82	±29	500
L4/L5	Left	809.61	±57	500
L5/S1	Right	824.24	±68	500
L5/S1	Left	862.98	±62	500

**Table 3 medsci-12-00034-t003:** Kappa Statistics Analysis of Interobserver and Intraobserver Agreement.

Agreement Level	Kappa Value Range	Agreement Definition	Interobserver Agreement	Intraobserver Agreement	Consensus Achieved
Poor	Less than 0.00	Disagreement beyond chance	Not Observed	Not Observed	No
Slight	0.00–0.20	Minimal agreement, mostly chance	Not Observed	Not Observed	No
Fair	0.21–0.40	Fair agreement, slightly beyond chance	Not Observed	Not Observed	No
Moderate	0.41–0.60	Moderate agreement, evident consensus	Not Observed	Not Observed	No
Substantial	0.61–0.80	Strong agreement, high level of consensus	Not Observed	Not Observed	No
Almost Perfect	0.81–1.00	Near-universal agreement, very high consensus	Yes	Yes	Yes

**Table 4 medsci-12-00034-t004:** Comparative Analysis of Lumbar Foraminal Volumetric Studies Using MRI and CT.

Criteria	MRI-Based Study Nurmukhametov et al.	CT-Based Study (Khiami et al.) [64]
**Study Objective**	To establish normative volumetric parameters of lumbar foramina.	To develop and validate a new method for measuring lumbar foraminal volume using CT.
**Measurement Tool**	High-resolution MRI.	CT scan with VitreaCore^®^ software for volume calculation.
**Participant Demographics**	500 patients, broad age range, gender-balanced.	10 healthy patients, mean age 26.3 years.
**Volumetric Results**	Detailed volumetric measurements from L1/L2 to L5/S1, showing progressive increases.	Average foraminal volumes at L4–L5 around 1.25 mm^3^ and 1.29 mm^3^ for two observers.
**Reproducibility**	High interobserver (ICC = 0.91) and intraobserver (ICC = 0.95) reliability.	High intraobserver correlations (up to 0.99) and interobserver correlations (up to 0.83).
**Reliability**	Demonstrated through ICC values indicating excellent consistency among measurements.	Demonstrated through intra- and interobserver correlation coefficients.
**Clinical Implications**	Provides baseline for normal variations, crucial for diagnosing and managing lumbar foraminal stenosis.	Suggests that CT-based measurements could supplement tools for measuring foraminal stenosis.

## Data Availability

The original contributions presented in the study are included in the article, further inquiries can be directed to the corresponding author/s.

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
