# Peer review of "Quantifying Lumbar Foraminal Volumetric Dimensions: Normative Data and Implications for Stenosis—Part 2 of a Comprehensive Series"

_medsci, 2024, doi:10.3390/medsci12030034_

Round 1
Reviewer 1 Report
Comments and Suggestions for Authors
This study presents a novel quantification of the volumetric dimensions of lumbar foramina across a significant patient cohort. The results demonstrate a clear trend of increasing foraminal volume from the L1/L2 to the L5/S1 levels, highlighting the spine's adaptive mechanisms to biomechanical demands and its capacity to accommodate the spatial requirements of nerve roots, particularly in the lower lumbar segments. The consistency of volumetric measurements, supported by strong interobserver and intraobserver agreements, underscores the reliability and reproducibility of the method. This study offers clinicians a precise tool for distinguishing pathological foraminal narrowing from normal anatomical variations by establishing quantifiable norms for lumbar foraminal volumes. I have a few comments that need to be addressed by the authors:
1. In Section “2.4 Patient Cohort”, the patient cohort is introduced. It is recommended to present the basic statistical information of the patients (maximum, minimum, mean) in a table.
2. In Sections 2.5 and 2.6, the inclusion and exclusion criteria are described respectively. It is suggested to use a flowchart to depict the selection process to enhance readability.
3. The font in Figure 3 is blurry, please replace it with a clearer image.
4. In Section “4.5.2 Enhanced Imaging Protocols”, you mention future exploration of artificial intelligence algorithms to improve the accuracy and reliability of the measurements. Volumetric measurements of the lumbar foramen are also part of medical diagnostics, involving the extraction of features from medical images and making predictions. Please consider adding references to medical AI diagnostic papers, such as “Enhanced Moth-flame Optimizer with Quasi-Reflection and Refraction Learning with Application to Image Segmentation and Medical Diagnosis” and “Generalized Oppositional Moth Flame Optimization with Crossover Strategy: An Approach for Medical Diagnosis” to enrich the paper's background.
Comments on the Quality of English Language
In Section “Abstract”, It is better to change "Material and method" to "Materials and Methods" for consistency with standard academic formatting.
Author Response
- In Section “2.4 Patient Cohort”, the patient cohort is introduced. It is recommended to present the basic statistical information of the patients (maximum, minimum, mean) in a table.
Responde 1 The table was provided
2. In Sections 2.5 and 2.6, the inclusion and exclusion criteria are described respectively. It is suggested to use a flowchart to depict the selection process to enhance readability.
Responde 2. The established was corrected
3. The font in Figure 3 is blurry, please replace it with a clearer image.
Responde 3: It got better
- In Section “4.5.2 Enhanced Imaging Protocols”, you mention future exploration of artificial intelligence algorithms to improve the accuracy and reliability of the measurements. Volumetric measurements of the lumbar foramen are also part of medical diagnostics, involving the extraction of features from medical images and making predictions. Please consider adding references to medical AI diagnostic papers, such as “Enhanced Moth-flame Optimizer with Quasi-Reflection and Refraction Learning with Application to Image Segmentation and Medical Diagnosis” and “Generalized Oppositional Moth Flame Optimization with Crossover Strategy: An Approach for Medical Diagnosis” to enrich the paper's background.
Responde 4: That section was improved according to your suggestion.

Reviewer 2 Report
Comments and Suggestions for Authors
Its an eccelent work.
I would like some more clear indication of level of help to the surgeon of spine.
Author Response
Thank you for your positive feedback and for recognizing the quality of our work. We appreciate your request for a clearer indication of how our volumetric analysis can assist spine surgeons. Below, we outline the practical applications and benefits of this methodology in surgical planning and intervention.
Preoperative Planning:
- Detailed Anatomy Visualization: Provides a detailed anatomical map for planning the safest and most effective surgical approach.
- Identification of Foraminal Stenosis: Identifies areas of significant nerve compression for targeted decompression surgery.
- Customized Surgical Approach: Tailors surgical plans to the individual’s specific anatomy and pathology, improving outcomes.
- Risk Stratification: Identifies high-risk cases for better preparation and resource allocation.
- Enhanced Communication: Facilitates better collaboration among the surgical team and improves patient education. the Part 3 of our work its on the way already where we suggest a classification of the lumbar foraminal stenosis
Reviewer 3 Report
Comments and Suggestions for Authors
Congratulations to the authors on the study on "Quantifying Lumbar Foraminal Volumetric Dimensions: Normative Data and Implications for Stenosis - Part 2 of a Comprehensive Series" it provides an in-depth analysis of lumbar foraminal volumes using high-resolution MRI data. However there are some pitfalls to be taken care of.
· Abstract is too long – please revise and shorten
· Please reduce the headlines in the materials and methods section (e.g. exclusion criteria) to make the manuscript more readable
· More in-depth description of the statistical methods required
· The description of the volumetric analysis lacks standardization, how was the depth determined? Standardized protocol?
· P10,line 405 - Interobserver and Intraobserver Agreement: Please elaborate – this is a crucial part of your study, hence simply referring to a table is insufficient.
· Revise table 2 – poor is mentioned twice
· Reformat table 3
· Overall reduction of headlines is recommended
· The manuscript should be checked again for typos and use of English, e.g.
· P4, Line 191: "terciary" should be "tertiary".
· P5, Line 218: "pathology" should be "pathologies"
Overall the study is scientifically sound and well-written.
Comments on the Quality of English LanguageCheck for typos - minor editing of use of English recommended
Author Response
Dear Reviewer,
Thank you for your constructive suggestions on our manuscript. We have made the following changes:
- Reduction of Headlines: We have reduced the number of headlines in the Materials and Methods section to improve readability.
- Detailed Statistical Methods: We have provided a more detailed description of the statistical methods used in the study, including volumetric analysis, correlation analysis, regression analysis, and reliability assessments.
- Standardization of Volumetric Analysis: We have standardized the measurement protocol, including the determination of depth using a standardized protocol and volume calculation.
- Elaboration on Agreement Analyses: We have elaborated on the interobserver and intraobserver agreement analyses, providing detailed explanations.
- Correction of Table 2: We have revised Table 2 to remove the duplicated "Poor" agreement level.
- Reformatted Table 3: We have reformatted Table 3 for better clarity and readability.
- General Typos and Use of English: We have reviewed the manuscript for typos and improved the use of English. Specific corrections include: Changed "terciary" to "tertiary" on page 4, line 191. Changed "pathology" to "pathologies" on page 5, line 218.
Thank you once again for your constructive feedback. We believe these revisions have strengthened our manuscript.
Round 2
Reviewer 3 Report
Comments and Suggestions for Authors
Congratulations to authors, they have significantly improved the quality of their manuscript. The manuscript should be checked again for typos - apart from that I happily recommend publication in its current state.
Comments on the Quality of English LanguageEnglish is fine, manuscript should be checked again for typos.
Author Response
Dear editor, here we provide you with the latest edition of the manuscript.
